# An Updated Review on Developing Small Molecule Kinase Inhibitors Using Computer-Aided Drug Design Approaches

**DOI:** 10.3390/ijms241813953

**Published:** 2023-09-11

**Authors:** Linwei Li, Songtao Liu, Bi Wang, Fei Liu, Shu Xu, Pirui Li, Yu Chen

**Affiliations:** 1Jiangsu Key Laboratory for the Research and Utilization of Plant Resources, Institute of Botany, Jiangsu Province and Chinese Academy of Sciences, Nanjing 210014, China; xt_llw@163.com (L.L.); 2021202069@stu.njau.edu.cn (S.L.); wangbi@cnbg.net (B.W.); liufei@cnbg.net (F.L.); xushu@cnbg.net (S.X.); 2Jiangsu Province Engineering Research Center of Eco-Cultivation and High-Value Utilization of Chines Medicinal Materials, Institute of Botany, Jiangsu Province and Chinese Academy of Sciences, Nanjing 210014, China; 3Key Laboratory of Pesticide, College of Plant Protection, Nanjing Agricultural University, Nanjing 210095, China

**Keywords:** CADD, SBDD, LBDD, kinase, small molecule kinase inhibitors

## Abstract

Small molecule kinase inhibitors (SMKIs) are of heightened interest in the field of drug research and development. There are 79 (as of July 2023) small molecule kinase inhibitors that have been approved by the FDA and hundreds of kinase inhibitor candidates in clinical trials that have shed light on the treatment of some major diseases. As an important strategy in drug design, computer-aided drug design (CADD) plays an indispensable role in the discovery of SMKIs. CADD methods such as docking, molecular dynamic, quantum mechanics/molecular mechanics, pharmacophore, virtual screening, and quantitative structure–activity relationship have been applied to the design and optimization of small molecule kinase inhibitors. In this review, we provide an overview of recent advances in CADD and SMKIs and the application of CADD in the discovery of SMKIs.

## 1. Introduction

Drug marketing is a costly and lengthy process. According to statistics, ninety percent of drugs that entered clinical trials failed to get FDA approval, mainly because of various problems encountered during rational drug design [1]. With the rapid development of bioinformatics and computer technology, computer-aided drug design (CADD) has made great progress. CADD can be divided into structure-based drug design (SBDD) and ligand-based drug design (LBDD) [2]. SBDD is based on the structure of the receptor and the ligand, the analysis and evaluation of the interaction between ligand and receptor, and the selection or modification of the structure of ligands to enhance affinity with the receptor [3]. LBDD can be used to obtain information such as molecular structure, charge distribution, and reactive site, which is helpful for understanding the relationship between the structural properties of compounds and their pharmacological activities or druggability [4]. CADD not only reduces the cost of drug research and development but also greatly shortens the time from discovery to market, hence it plays an important role in drug research and development [5].

Kinases play an important role in regulating cell metabolism, growth, exercise, differentiation, and division, as well as signal pathways associated with the formation and development of many human diseases, including cancer [6,7,8,9], vascular diseases [10,11,12,13], diabetes [14,15,16], inflammation [17,18,19,20,21], and degenerative diseases [22,23,24], which makes them attractive targets for drug discovery. Since Imatinib was approved in 2002, the US Food and Drug Administration (FDA) has approved 79 small molecule kinase inhibitors (SMKIs), and there are currently several small molecule kinase inhibitor candidates in clinical trials. Most kinase inhibitors bind to the kinase catalytic domain, that is, the ATP-binding site. Due to the high conservation of the ATP-binding site in kinases, off-target effects (i.e., low selectivity) of kinase inhibitors can result in undesirable side effects [25,26]. Therefore, finding novel, effective, and selective kinase inhibitors is still an important challenge. CADD can be used to analyze the subtle differences and structural characteristics of different kinase ATP-binding sites and promote the discovery of kinase inhibitors. As of July 2023, a total of 25 kinase inhibitors discovered using the CADD method have been approved by the FDA.

In this review, we provide an overview of recent advances in CADD application, including structure-based drug design (SBDD) and ligand-based drug design (LBDD) methods for drug discovery. Moreover, we focus on the recent progress in kinase inhibitors and highlight representative investigations on kinase inhibitors using CADD.

## 2. Computer-Aided Drug Design

Traditional drug research and development (R&D) uses random screening methods. The average cost of drug development from laboratory research to market launch is about 2.558 billion US dollars, lasts at least 13.5 years, and has a success rate of only about 10.0% [27,28]. To meet the needs and accelerate the process of new drug R&D, the concept of drug molecular design has been proposed, propelling drug investigation into a new era. Drug molecular design is a scientific and efficient strategy that reasonably designs new chemical entity molecules with expected pharmacological activities, safety, stability, and quality control properties based on the structural characteristics of endogenous ligands and targets [29].

Computer-aided drug design (CADD) provides new ideas for rational drug molecular design and promotes significant breakthroughs in drug molecular design. In 1981, the cover article of Fortune “Next Industrial Revolution: Designing Drugs by Computer at Merck” marked the formal entry of CADD into the field of pharmaceutical research [30]. Subsequently, the Nobel Prize in Chemistry in 1998 and 2013 was awarded for related work in molecular simulation, demonstrating that experiments and theory are twin pillars in chemistry. In the past few decades, scientists have developed a variety of CADD methods, including sequence alignment [31], homologous modeling [32], molecular docking [33], pharmacophore [34], quantitative structure–activity relationship [35], molecular dynamics [36], and quantum mechanics [37]. These methods have been adopted at different stages of drug development. Compared with traditional drug R&D methods, CADD reduces the high cost, decreases the high-risk problems, shortens the cycle time, and improves the efficiency of new drug development (Figure 1). Due to its remarkable advantages, CADD has been used to successfully develop many drugs, including Donepizil (treatment of Alzheimer’s disease), Zanamivir (anti-virus), Imatinib (anti-tumor), Saquinavir (treatment of immunodeficiency virus) and so on [38,39,40].

Computer-aided drug design [1,2,3] includes structure-based drug design (SBDD) and ligand-based drug design (LBDD) (Figure 2). The prerequisite for the implementation of SBDD and LBDD is the 3D structure of targets and molecular structures of ligands, respectively. The structure-based drug design method uses the three-dimensional structure of biological targets to find hit or lead compounds using ab initio design, molecular docking, or virtual screening methods by measuring the interaction modes and binding energy between ligand and receptor [2]. Ligand-based drug design is used to perform quantitative structure–activity relationship and pharmacophore modeling to establish the theoretical prediction model between molecular structure and biological activity, which is used for screening and optimizing active compounds [3]. Given the advantages and disadvantages of SBDD and LBDD, researchers typically use a combination of SBDD and LBDD methods for drug design to achieve better results.

### 2.1. Structure-Based Drug Design

#### 2.1.1. Preparation of Target Structures

To achieve rational structure-based drug design, it is necessary to fully understand the three-dimensional structure of targets. Structural biology is devoted to studying the complete, accurate, and real-time three-dimensional structure of biological macromolecules via experimental means [41,42]. The structures of a large number of biological macromolecules have been resolved using structural biology technology. Structural data of biological macromolecules are collected in the Protein Structure Database (PDB, https://://www.rcsb.org, accessed on 16 July 2023). As of July 2023, information on 207,540 biological macromolecular structures has been collected in the PDB database and the number of structures is increasing rapidly—by nearly 10,000 pieces per year (Figure 3). The structures of macromolecules include a variety of drug targets associated with major human diseases, including antitumor [43], antibacterial [44], antiviral [45], Alzheimer’s disease [46], and diabetes [47].

However, due to the limitations of experimental techniques for structural analysis and the inherent properties of biological macromolecules, there are still a large number of drug target structures that are yet to be accurately analyzed. With the help of computer simulation technology, scientists have developed a series of molecular structural prediction methods. The traditional method of simulating the structure of targets involves a combination of homology modeling [32] and molecular dynamics (MD) [36] (Figure 4). First, based on the principle that the primary structure of a protein determines its tertiary structure, homology modeling constructs the three-dimensional structure of the protein through sequence alignment, structural construction (copying skeleton, constructing side chains, completing missing residues, and optimizing loop regions), and energy minimization. Then, molecular dynamics technology simulates the nearly real and accurate structure of protein structures obtained from homology modeling in a solvent state. Moreover, many software for protein prediction have also been developed in artificial intelligence, including AlphaFold and RosettaFold.

The widespread application of the above scientific and technological advancements contributes to the deep understanding of the molecular mechanisms underlying the occurrence and development of related diseases and provides new perspectives and research tools for rational structure-based drug design. The basic process of structure-based drug design is: (1) tailor-made ligand molecules with shape matching, electrical complementarity, and clear intermolecular interactions are designed at the molecular level based on the three-dimensional structural characteristics of the target, (2) the structure of the ligand–receptor complex is established by molecular simulation to verify the completion of drug design, (3) a combined analysis of the binding energy between ligand and receptor, the physical and chemical properties of ligands, and biological activity is conducted to evaluate the activity and druglikeness of designed compounds; this guides subsequent drug design and optimization (Figure 5). Structure-based drug design greatly reduces the blindness of drug design and screening and accelerates the research and development process of lead compounds and candidate drugs. Among the SBDD methods, molecular docking, molecular dynamics (MD), and quantum chemistry (QM) are commonly used to simulate and analyze the structure of ligand–receptor complexes, predict the binding strength between ligand and receptor, and calculate the physical and chemical properties of compounds. The following sections provide a detailed introduction to the above three methods.

#### 2.1.2. Molecular Docking

Molecular docking has gradually developed into the most frequently used method since the early 1980s, with high success rates in structure-based drug design (Table 1) [48,49,50,51]. Molecular docking predicts the orientation, binding pose, and binding energy between small molecular ligands and the active sites of targets at the atomic level, and the optimal binding pose is screened via comprehensive consideration of space-matching and energy-matching to clarify the ligand–receptor interaction [52,53,54]. Graphic analysis software can be used to visualize the ligand–receptor binding mode and observe the fine structural differences between homologous targets, offering key microscopic features for the rational design of compounds with high target compatibility and selectivity.

Scientific research achievements using molecular docking for drug design have increased year by year in the past three decades, covering almost all disease fields in pharmaceutical research, including anti-tumor [55], anti-virus [56], anti-infection [57], diabetes [58], anti-hypertension [59], anti-depressant [60], anti-inflammatory [61], and anti-bacterial [62] research. Undoubtedly, molecular docking has become a necessary tool in the research and development of targeted drugs and is one of the most widely used and effective means of designing and screening hit-and-lead compounds [63,64,65].

**Table 1 ijms-24-13953-t001:** Popular molecular docking software.

Software	Algorithm	Scoring Function	Website	Ref.
Dock	Fragment growth	Force field, Surface matching score, Environment matching score	http://dock.compbio.ucsf.edu/DOCK_6/, accessed on 16 July 2023).	[66]
AutoDock	Genetic algorithm	Environment matching score	http://autodock.scripps.edu/, accessed on 16 July 2023).	[67]
GOLD	Genetic algorithm	Empirical	http://www.biosolveit.de/FlexX/, accessed on 16 July 2023).	[68]
FlexX	Fragment growth	Empirical	https://github.com/flexxui/flexx, accessed on 16 July 2023).	[69]
Z-Dock	Geometric matching/Molecular dynamics	CAPRI+	http://zdock.umassmed.edu/, accessed on 16 July 2023).	[70]
Hex	Geometric matching	CAPRI+	http://www.csd.abdn.ac.uk/hex/, accessed on 16 July 2023).	[71]
SLIDE	Systematic	Force field, Empirical	http://www.bmb.msu.edu/~kuhn/software/slide/, accessed on 16 July 2023).	[72]
Fred	Systematic	Empirical	http://www.eyesopen.com/oedocking, accessed on 16 July 2023).	[73]
LeDock	Annealing–Genetic algorithm	Physics/knowledge hybrid	http://www.lephar.com/software.htm, accessed on 16 July 2023).	[74]
Glide	Systematic	XP/SP/HTVS	https://www.schrodinger.com, accessed on 16 July 2023).	[75]
Surflex-Dock	Hammerhead	Empirical	http://www.tripos.com, accessed on 16 July 2023).	[76]

#### 2.1.3. Molecular Dynamics

Drug-target binding is a competitive process involving ligands, receptors, and solvents. The solvent environment in which ligands and receptors are located plays an important role in molecular recognition and binding energy calculation. The solvent effect is not considered in molecular docking, making it impossible to provide accurate microscopic information on ligand–receptor interactions. Furthermore, cell life activity is accompanied by protein conformational changes, thus elucidating the details of drug-target protein conformational changes is helpful in designing efficient and highly selective compounds. However, at present, the dynamic process of protein conformational change cannot be measured using experimental methods.

Based on Newton’s laws of motion, MD uses a computer to simulate the motion state of molecules and atoms in a certain period, measuring the behavior of system properties and other parameters with time from a dynamic perspective [77,78,79,80]. Molecular dynamics can be used to observe many details at the microscopic level that cannot be captured in real experiments, for example, potential drug-binding sites. The huge simulation system, long-time simulation, and high-precision calculation method make the cost of molecular dynamics simulations expensive (Table 2). The sample count for MD is usually small to reduce the calculation cost of MD. MD is often used to assist in the initial and post-processing stages of molecular docking, which involves three aspects: protein flexibility, docking complex refinement, and binding energy calculation (Figure 6) [81,82,83,84,85,86].

In the initial stage of molecular docking, molecular dynamics simulation mainly performs the work of “protein flexibility”. Different conformers of the target are extracted from the molecular dynamics simulation trajectory using “snapshots” as the initial structures for multiple receptor conformations (MRC) molecular docking, which nearly truthfully simulates the existing form of the target under physiological conditions. The Merck company discovered the “mysterious binding pocket” of HIV integrase through MD simulation and then discovered the highly potent antiviral drug Raltegravir by structure-based drug design targeting of this site [94]. In the post-processing stage of docking, MD mainly performs two functions: “docking complex refinement” and “binding energy calculation”. The preferential conformational structure simulated by molecular docking was refined by considering the influence of solvent, complex flexibility, and induced fit to obtain accurate ligand–receptor complex structure and interaction between ligand and receptor. The molecular dynamic-induced fit (MD-IF) strategy has been widely used to refine the sub-binding pocket of active sites, find allosteric regulatory sites, and improve the binding mode of ligand–receptor complexes [81,82,83,84,85,86,95,96,97]. The sophisticated intermolecular interactions between the HIV-1 protease inhibitor Darunavir [98], the Janus kinase 2 (JAK2) kinase inhibitor Ruxolitinib [99], the TAF1 protein second bromine domain inhibitor acetylated lysine [100], and corresponding targets were predicted using MD-IF simulation. As for the ligand–receptor complex, MD simulation considers the influence of dominant solvents, especially the influence of water molecules in macromolecular structure and key water molecules near the binding site, which is critical for the accurate calculation of binding energy between receptor and ligand [101]. The structure of biomacromolecules and receptor–ligand interactions obtained using MD provide important clues for drug design and discovery; thus, MD has become one of the pivotal methods of structure-based drug design.

#### 2.1.4. Quantum Chemistry

Quantum chemistry (QC) is a method based on the basic principles of quantum mechanics (QM). It reveals the interaction, transformation, structure, and relationship between molecules in the chemical field by solving the Schrödinger equation and describing the motion of microscopic particles [102]. At present, a relatively complete theoretical system and several calculation methods have been established in quantum chemistry, such as ab inito methods (ab initio, HF), density functional theory (DFT), and semi-empirical methods (PM3, AM1, PM6, etc.).

Since quantum chemistry calculation involves the interaction between electrons, it plays an irreplaceable role in many fields such as life science, medical science, material science, and so on [103,104,105,106,107]. It breaks the boundaries between different disciplines and promotes interdisciplinary collaboration. For example, quantum medicinal chemistry [108] plays a vital role in the discovery and development of innovative drugs and is mainly used to explore the structures and properties of drug molecules, the interaction between biological macromolecules and ligands, and chemical reaction mechanisms. The drug targets are usually macromolecules, such as proteins, receptors, and nucleic acids. For such large systems, DTF and the ab initio method need a lot of calculation time when dealing with a large number of electron correlations, and the semi-empirical method has poor accuracy. To overcome the above shortcomings, Martin Karplus et al. proposed the hybrid strategy of quantum mechanics and molecular mechanics (QM/MM) for the simulation of macromolecular systems in the 1990s. It has the characteristics of high calculation accuracy and fast calculation speed. Three scientists, including Martin Karplus, were awarded the 2013 Nobel Prize in Chemistry for their contribution to the development of QM/MM [109,110,111,112,113]. The basic algorithm of the QM/MM method involves dividing the whole molecular system into two layers. The core region of the study system (such as the active cavity in the ligand–receptor complex) is treated using the QM method, while the peripheral region (such as the inactive cavity in the ligand–receptor complex and the solvent environment) is treated using the MM method. The QM/MM simulation model generates calculated results for the entire system that are similar to those obtained by the QM calculation method owning to the accurate consideration of changes in electronic structure [114,115,116,117,118].

QM/MM has gradually become a powerful tool in biology, crystal engineering, material science, and supramolecule fields. In drug design, QM/MM is used to refine the structure of ligand–receptor complexes, calculate the binding energy between ligand and receptor, predict physical and chemical properties (charge density, pKa, water solubility, etc.) of ligands, and simulate the action of endogenous ligands, drug metabolism pathway, and drug resistance mechanism (Figure 7) [119,120,121,122]. In summary, QM/MM provides more refined structural and property parameters for structure-based drug design and has been used to assist in the study of drug targets such as tubulin [123], human carbonic anhydrase (hCAII) [124], cyclin-dependent kinase 2 (CDK2) [125], and lipoxygenase [126].

#### 2.1.5. Molecular Docking–Molecular Dynamics–Quantum Chemistry

Molecular docking, molecular dynamics, and quantum chemistry are the three methods that are commonly used in structure-based drug design. These methods have their own advantages and disadvantages. Generally speaking, molecular docking can be used to quickly obtain the dominant conformer of the ligand–receptor complex and provide a preliminary analysis of the interaction between them. Molecular docking has the fastest calculation speed but the lowest calculation accuracy of the three methods. Molecular dynamics can provide the dynamic process of molecular motion over time at the microscopic level. Although it can realistically simulate the dynamic process of macromolecular systems in solution, molecular interactions cannot be accurately described due to the use of molecular force fields in the MD algorithm. Not only does the QM/MM hybrid method have high computational speed, but it also has high computational accuracy. QM/MM can be used to predict the structure, properties, binding energy, and interaction of ligand–receptor complexes accurately, but not for obtaining the global dynamic conformers of ligand–receptor complexes. The three mentioned computation methods can complement each other, enhancing the strengths and circumventing the weaknesses. A combination of docking, MD, and QM/MM is generally used to simulate structures and properties of receptors and ligands for SBDD. In terms of structure, it is possible to distinguish subtle differences between homologous receptors, obtain accurate structures of ligand–receptor complexes, and reasonably analyze fine interactions between the ligand and receptor using the combined calculation method. As for properties, the combined method (docking–MD–QM/MM) can estimate the activity of small molecular drugs by calculating ligand–receptor binding energy and evaluate druglinkeness by predicting the metabolic site, lipid–water partition coefficient, atomic charge, and other properties of the given molecule. A variety of drug screening platforms have been established based on the combination of multi-molecular simulation technologies, including the MD high-throughput screening–SBDD–MD activity testing module built by Caflisch’s research group [127] has been applied to design and screen many targeted drugs efficiently and accurately. Ahmed et al. used a combination of docking, MD, and QM/MM to screen for HER2 inhibitors (Figure 8). First, **M8** and **M19** were screened from the designed compounds by docking scoring. MD simulation and QM/MM calculation of **M8** and **M19** were then performed to evaluate their biological activity. Finally, **M8** and **M19** demonstrated anti-HER2 activity *in vitro*, which corresponded well with simulated data [128].

#### 2.1.6. Virtual Screening

Virtual screening (VS) technology is a method of screening potential active compounds by evaluating the binding energy between compounds and targets in small molecule databases [129]. Molecular docking is ordinarily used in the VS process, thus VS is a rapid and cheap drug screening method. The commonly used databases of small molecular compounds are Zinc (https://zinc.docking.org/, accessed on 16 July 2023), DrugBank (https://www.drugbank.com/, accessed on 16 July 2023), PubChem (https://pubchem.ncbi.nlm.nih.gov/, accessed on 16 July 2023), ChEMBL (https://pubchem.ncbi.nlm.nih.gov/source/ChEMBL, accessed on 16 July 2023), and BindingDB (https://www.bindingdb.org/, accessed on 16 July 2023). Virtual screening has become a common method in drug screening, especially in screening hit-or-lead with novel skeletons. Its great potential and value have been confirmed by the discovery of various drugs such as cyclin-dependent kinase (CDK) inhibitors [130], chitinolytic enzyme inhibitors [131], and anti-coronavirus SARA-CoV-2 compounds [132] (Figure 9). For example, Yuan et al. screened a lead compound as a CDK inhibitor from an in-house database; further chemical optimization led to the highly selective and potent CDK4/6 inhibitor.

As a rational, scientific, and efficient drug design method, structure-based drug design plays an important role in the discovery of hit compounds, optimization of lead compounds, and development of innovative drugs. Since Captopril—the first drug discovered by SBDD—came on the market, the anti-HIV drugs Raltegravir and Amprenavir, the anti-tumor drugs Imatinib and Ponatinib, the anti-coagulant drugs Rivaroxaban and Dabigatran etexilate, the anti-viral drugs Oseltamivir and Boceprevir, and many other blockbuster drugs have come out one after another. As of July 2023, a total of over 70 drugs discovered using SBDD have been marketed, and a large number of excellent candidate drugs are in clinical and preclinical trials. Structure-based drug design has become a powerful means of designing, researching, and developing innovative drugs in the field of medicinal chemistry. In addition, diversity-oriented synthesis (DOS) [133], combinatorial chemistry [134], computer-aided synthesis planning (CASP) [135], artificial intelligence (AI) [136,137,138,139], robot systems, and other technologies have been used to explore novel chemical reactions, which has increased the number of synthetic blocks in organic synthesis, broadened the space of the chemical universe, and improved the possibility of pharmaceutical chemists to design a variety of ligand molecules. Therefore, we are bound to believe that SBDD will play an important role in drug research and development and continue to promote the process of novel drug discovery.

### 2.2. Ligand-Based Drug Design

#### 2.2.1. Quantitative Structure–Activity Relationship

In 2002, at the 70th anniversary of the founding of the Chinese Chemical Society, Academician Guangxian Xu proposed “the quantitative relationship between molecular structure and its properties” as one of the “four century puzzles in chemistry” in the 21st century. According to the basic law of chemistry, “structure determines performance, and performance reflects structure”, hence there must be a close relationship between the structure of a substance and its performance, and exploring the relationship between structure and performance has become one of the hot topics in chemical research.

Quantitative structure–activity relationship (QSAR) is the main method of exploring the relationship between the structures and properties of compounds. The theoretical basis of this method is that the structural characteristics of a series of compounds with similar structures and the same mechanism of action are correlated with their activities/properties (Figure 10) [140,141,142]. In a broad sense, QSAR activity includes physiological and biochemical indexes (e.g., pharmacological activity, enzyme inhibition toxicity, and neurotoxicity) and physical and chemical properties (e.g., solubility, retention time, lipid–water partition coefficient, reaction rate constant, boiling point, and melting point) [141]. In pharmaceutical research, QSAR research follows the following steps: first, a mathematical model of the structures and activities/properties of known or assumed drug molecules or active compounds is generated using statistical methods to reveal the mechanism and mode of action of drug molecules; the obtained mathematical model is then used to quantitatively explain and predict the activities and properties of unknown compounds, to optimize and design drug molecules rationally and effectively. At present, QSAR is widely used in the pharmaceutical field for the prediction of the pharmacological activity of compounds, discovery and optimization of lead compounds, and evaluation of absorption, distribution, metabolism, excretion, and toxicity (ADMET) and other drug properties [142,143,144,145,146,147].

QSAR is one of the earliest and most widely used strategies in the ligand-based drug design field. Its characteristics include a small amount of calculation, high accuracy, and a short calculation period. It plays an important role in predicting and screening compounds and is especially suitable for rational drug design when the target structure is unknown. Quinolone antibiotics, monoamine oxidase (MAO) inhibitors, HIV-1 integrase inhibitors, proteolytic enzyme inhibitors, tyrosinase inhibitors, epidermal growth factor receptor (EGFR) kinase inhibitors, and acetylcholinesterase (AChE)/butyryl-cholinesterase (BuChE) enzyme inhibitors were discovered using the QSAR method (Figure 11) [148,149,150,151]. For example, Hajalsiddig et al. established a 2D-QSAR model of EGFR inhibitors, and a series of new compounds were then designed and evaluated based on QSAR-derived information.

#### 2.2.2. DFT-Based Quantitative Structure–Activity Relationship

The primary role of QSAR research is to select appropriate and sufficient molecular structural descriptors that are crucial for establishing a good QSAR model. The ideal descriptors can accurately and perfectly determine the structural information that affects biological activity, hence the established QASR model has not only high predictive ability, it also has clear physical meaning that can be used to explain the mechanism of action of compounds.

Commonly used molecular descriptors include molecular surface area, molecular volume, lipid–water partition coefficient, Hammett electrical parameters, thermodynamic parameters, and stereoscopic effect parameters (Table 3). The above descriptors are measured experimentally or obtained by fitting empirical parameters, which usually show the shortcomings of a large experimental workload and inaccurate parameter data. The quantum chemistry parameters calculated using quantum chemistry calculations have become an important approach for obtaining molecular structure descriptors in QSAR research. Compared with traditional molecular descriptors, quantum chemical parameters demonstrate three advantages: first, quantum chemical parameters are more accurate and have clear physical and chemical significance; second, quantum chemical parameters can be fully and efficiently simulated theoretically, without experimental measurements; third, quantum chemistry parameters are not limited to obtained compounds, but can also evaluate designed compounds in advance [152,153]. Thence, quantum chemical parameters have gradually penetrated the field of QSAR research, particularly density functional theory-based quantitative structure–activity relationship (DFT-based QSAR) [154]. DFT-based QSAR has been widely used in the development stage of various types of drugs, including adrenergic receptor inhibitors [155], calcium ion channel blockers [156], fatty acid synthase inhibitors [157], protoporphyrinogen oxidase inhibitors [158], and melanin-concentrating hormone receptor 1 inhibitors [159]. In the study of kinase inhibitors, DFT-QSAR has been applied to mammalian target of rapamycin (mTOR) kinase inhibitors [160], phosphatidylinositol 3 kinase (PI3K) inhibitors [161], and cyclin-dependent kinase (CDK) inhibitors [162].

#### 2.2.3. Pharmacophore Modeling

In 1909, Ehrlich put forward the concept of pharmacophore, that is, the key structural features of compounds determine their biological activity. The earliest-developed pharmacophore method is ligand-based pharmacophore, which aims to identify the common structural characteristics of compounds with similar pharmacological activities and different structures [163,164,165,166]. Pharmacophore is an important strategy for CADD research when the target structure is unknown. The commonly used automatic pharmacophore generation programs include Discovery Studio, PHASE, LigandScout, and MOE. BCR-ABL tyrosine kinase inhibitors [167], succinate dehydrogenase inhibitors [168], and acetylcholinesterase inhibitors [169] were discovered using the pharmacophore method (Figure 12).

#### 2.2.4. Molecular Similarity

Molecular similarity refers to the structural similarity between two compounds, which is an important application of the chemoinformatics method in CADD. The basic idea is that compounds with similar structures may have similar pharmaceutical activity. Drug screening based on molecular similarity involves screening out compounds with structural similarity from a compound database using active compounds as templates. Molecular similarity is not only an approach for screening hit-or-lead compounds but also a guide for the optimization of compounds [170,171]. For example, the principle of the electron-isosteric substitution drug design strategy is based on molecular similarity.

## 3. Kinases

Kinases participate in many important physiological processes in organisms and an increasing number of signaling pathways have been confirmed to be closely associated with substrate phosphorylation. The 1989 Nobel Prize was awarded to Bishop and Varmus for their discovery of protein kinases associated with tumors [172]. The 1992 Nobel Prize in Medicine was awarded to Fischer and Krebs for the discovery that the reversible protein phosphorylation process is a biological self-regulation mechanism and the imbalance in intracellular substances can lead to the occurrence of diseases [173]. The 2001 Nobel Prize was awarded to Nurse and Hunt for their discovery of the important regulatory role of cell cycle-dependent protein kinases in the cell cycle [174]. Numerous investigations have shown that substrate phosphorylation, kinases, and their regulatory mechanisms play an important role in the field of life sciences.

Reversible phosphorylation reactions can affect and regulate various functional cell processes by regulating and balancing the activity of substrates, for example, protein synthesis, gene expression, signal factor release, cell metabolism, morphological changes, and apoptosis. Abnormal phosphorylation levels and function are closely related to many diseases such as tumors, inflammation, immunity, cardiovascular diseases, neurodegenerative diseases, and diabetes. Kinases have become one of the important targets in drug research and development and kinase inhibitors have become drugs for treating various important diseases (Figure 13). Based on reports in the literature, about one-third of drugs under research or development worldwide are associated with kinases.

### 3.1. Structure and Function of Kinases

The general structure of kinases is shown in Figure 14. It includes a catalytic domain, a helical domain, and multiple binding domains. The function of kinases is to phosphorylate substrates, thus its catalytic domain is highly conserved, and the three-dimensional structures of different kinases are very similar. The catalytic domain is composed of a bilobate region linked by a hinge region, a G-loop region (also known as P-loop), a catalytic loop, and an activation loop (A-segment). Bilobate regions are located at the *C-* and *N*-termini of kinases, called *C*-lobe and *N*-lobe fragments, respectively. The *N*-lobe is composed of five *β*-sheets and an *α*-helix, while the *C*-lobe is mainly composed of α-helix structures. The catalytic cleft between the bilobate region is an ATP binding site and is also the binding site for most ATP-competitive kinase inhibitors [175,176,177].

Kinases exist in two states—active and inactive—that act as switches during the signal transduction process (Figure 15). Under normal physiological conditions, most kinases exist in the inactive state and when activated by upstream signals, they initiate the substrate phosphorylation reaction. The Asp-Phe-Gly motif (DFG motif) located at the *N*-terminal of the activation loop is recognized as playing an important role in regulating kinase activity. The conformation of amino acid residues in the DFG motif characterizes different kinase states [178,179,180]. When kinases are in the inactive state, the aspartate residue in the DFG motif faces away from the ATP binding site, leading to the activation loop preventing the substrate from coming into contact with the active site, that is, the “DFG out” conformation. When the kinase is in the active state, the conformation of the DFG motif is reversed and the aspartic acid residue faces the ATP binding site, enabling the kinase to function normally, namely, the “DFG-in” conformation [181].

The mechanism of substrate phosphorylation that is catalyzed by kinases [182] is shown in Figure 16. Phosphatidylinositol 3 kinase (PI3K) is taken as an example to describe the phosphorylation of the substrate: first, ATP and substrate phosphatidylinositol 4,5-biphosphate (PIP_2_) bind to the catalytic domain of PI3K, and the activation loop, phosphate groups of ATP, and Mg^2+^ ions form a molecular interaction; His^936^ then captures protons from PIP_2_, allowing PIP_2_ to act as a nucleophile and attack the *γ*-phosphoryl group; finally, the phosphoryl group is transferred to PIP_2_ to generate phosphatidylinositol-3,4,5-triphosphate (PIP_3_), thus completing the phosphorylation process.

Based on their substrate, kinases can be divided into protein kinases, lipid kinases, carbohydrate kinases, and other kinases. Protein kinases consist of eight categories: AGC (e.g., PKA, PKC, and PKG), CMGC (e.g., CDK, CDKL, MAPK, and CLK), CAMK (e.g., CaMKI, MLCK, and eEF2K), CK1 (e.g., CK1, TTBK, and VRK), STE (e.g., STE20, STE11, and MAP4K), TK (e.g., RTK, and CTK), TKL (e.g., MLK, MLKL, and RAF), and atypical kinases (e.g., ATR, mTOR, and ADCK1). The most common lipid kinase is PI3K. Currently, there are over one million studies associated with kinases that have been published, over 500 kinase-related protein structures that have been identified, and over 900 kinase sequences that have been analyzed. Additionally, over four-fifths of human kinase groups have been determined [183].

### 3.2. Small Molecule Kinase Inhibitors

Studies have shown that kinases are closely associated with human diseases, thus more pharmaceutical companies and research institutions are focusing on developing kinase inhibitors. Kinase inhibitors include macromolecule antibody inhibitors and small molecule kinase inhibitors (SMKIs). SMKIs have several advantages, including oral administration, easy operation, and low cost compared with antibody inhibitors, thus SMKIs have become one of the hotspots of targeted drug research and development [184].

Based on the mode of binding between small molecule kinase inhibitors and their targets, SMKIs can be divided into reversible SMKIs and irreversible SMKIs. Irreversible SMKIs occupy the ATP-binding site by forming covalent bonds with residues and then sealing the ATP-binding pocket, blocking kinase function. Reversible SMKIs can be divided into four types based on ligand binding sites and the conformation of the DFG motif (Figure 17). Type I inhibitors are ATP-competitive inhibitors that bind to the active conformation of the kinase, with the aspartate residue (white backbone) of the DFG motif pointing into the ATP-binding pocket [185]; type II inhibitors bind and stabilize the inactive conformation of the kinase, with the flipped aspartate residue facing outward from the binding pocket [186]; type III inhibitors occupy an allosteric pocket that is adjacent to the ATP-binding pocket but does not overlap with it [187]; type IV inhibitors bind to an allosteric pocket remote from the ATP-binding pocket [188].

In the 1950s, researchers realized that kinase inhibitors play a crucial role in signal transduction and started theoretical investigations of kinase inhibitors. In the 1980s, the studies and applications of epidermal growth factor receptor (EGFR) inhibitors opened a new chapter in the study of kinase inhibitors [189,190]. Imatinib (Gleevc^®^), the first kinase inhibitor, was approved by the US Food and Drug Administration (FDA) in 2001, marking a milestone in the research of kinase inhibitors [191]. In the ensuing 20 years, over 70 kinase inhibitors were approved by the FDA (Figure 18),[192] and hundreds of SMKIs are currently in preclinical and clinical trials studying their effectiveness in the treatment of tumors, rheumatoid diseases, diabetes, and so on. Of the marketed SMKIs, the sales of over 10 drugs reached or exceeded $1 billion in 2022, indicating that small molecule kinase inhibitors are and will continue to be an important component of drug research and development.

## 4. Small Molecule Kinase Inhibitors Discovered Using CADD

Kinases play key regulatory roles in biological processes, including cell growth, differentiation, infiltration, angiogenesis, and metastasis. There are at least 500 different kinases in the human body, and it was initially believed that kinases would not be good drug targets due to the highly conserved ATP-binding pockets in kinases that would make it a challenge to design specific kinase inhibitors [25,26]. Computer-aided drug design strategy can distinguish the subtle differences and structural characteristics of different kinase ATP-binding sites and promote the discovery of kinase inhibitors. Imatinib, an Abl inhibitor, was the first kinase inhibitor discovered using SBDD in 2001. As of July 2023, a total of 25 kinase inhibitors had been approved by the FDA and hundreds of compounds under clinical investigation had been discovered using the CADD method (Figure 19). Furthermore, the proportion of marketed kinase inhibitors discovered using guided CADD has increased year-on-year. An overview of FDA-approved kinase inhibitors discovered using the CADD approach since 2021 is provided below (Figure 20, Figure 21, Figure 22 and Figure 23) [193,194,195,196,197,198,199].

To get a better understanding of the use of CADD in the discovery of kinase inhibitors, Deucravacitinib, Adagrasib, and Pirtobrutinib were selected to offer a detailed description. 

Deucravacitinib, a TYK2 inhibitor, was discovered by the Bristol-Myers Squibb Company and approved by the FDA in 2022 (Figure 20). During the Deucravacitinib design process, the sulfone group was first used as the lead compound (instead of the amide group) to break the interaction with conserved water and to form a H-bond with the target; second, a triazole fragment was introduced to the structure to improve potency and metabolic stability; the aromatic ring was then modified to form hydrogen bonds with the target; finally, the TYK2 inhibitor Deucravacitinib was obtained [197].

Adagrasib, a KRAS inhibitor, was approved by the FDA in 2022 (Figure 21). The investigator discovered the lead compound—which showed moderate inhibition against KRAS, with poor pharmacokinetic properties—in-house. To improve the pharmacokinetic properties of the lead compound, the investigator removed the hydroxyl group from the naphthalene ring to weaken *O*-atom metabolism. The cyanide group was then introduced into the piperazine ring to form hydrogen bonds with the amino acid residue of KRAS. A chlorine atom was subsequently added to the naphthalene ring and embedded in the hydrophobic cleft. Finally, an electrophile was introduced to the olefinic bond to attenuate reactivity [198].

The Pirtobrutinib discovery process is shown in Figure 22. Ibrtinib was selected as the lead compound. It contains three pharmacophore fragments, a hinge region, H3 pocket (hydrophobic pocket), and a covalent binding region (covalently binding with the cysteine at position 481). Optimization of the three regions was conducted. The pyrazolopyrimidine of Ibtinib was broken to obtain a primary amide with an amino group in the ortho position of pyrazole, and the two substituent groups formed a pseudo-bicyclic ring through hydrogen bonding; the benzene ring connected by oxygen in the H3 pocket was changed into an aromatic ring connected by amide methylene; the covalent binding region was changed to a non-covalently bonded CF_3_-substituted ethyl group; finally, Pirtobrutinib was optimized [199].

## 5. Future Perspectives

Despite the impressive achievements of small molecule kinase inhibitors, further research on kinases and kinase inhibitors is still needed. For kinases, the detailed function of many kinases needs to be clarified. Second, only one-fifth of human kinase targets have been reported with corresponding small molecule kinase inhibitors; thus, the druggability of other kinases needs to be investigated. For the SMKIs concept, the selectivity and resistance of SMKIs are key concerns; the available SMKIs are mainly type I and type II inhibitors, and most of them are protein kinase inhibitors. The development of novel SMKIs cannot be underestimated. Structure-based drug design can be based on target structure for rational drug design; ligand-based drug design can be used to extract the structural characteristics of active molecules. Computer-aided drug design and comprehensive consideration of ligand and receptor information greatly reduce the blindness of the experiment and improve the screening efficiency of lead compounds and candidates, thus saving manpower, material, and financial resources and shortening the drug development cycle. Given the widespread use of CADD in the development of kinase inhibitors, we believe that CADD will continue to promote kinase inhibitor research.

## Figures and Tables

**Figure 1 ijms-24-13953-f001:**
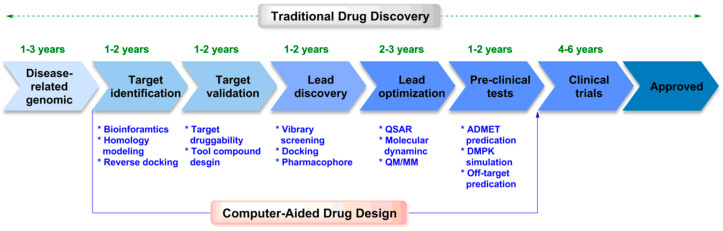
Comparison of traditional drug discovery and CADD in the drug discovery process.

**Figure 2 ijms-24-13953-f002:**
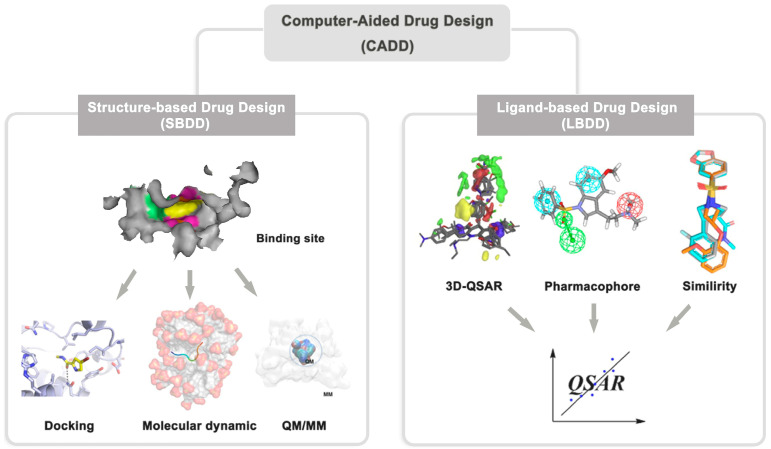
Description of computer-aided drug design.

**Figure 3 ijms-24-13953-f003:**
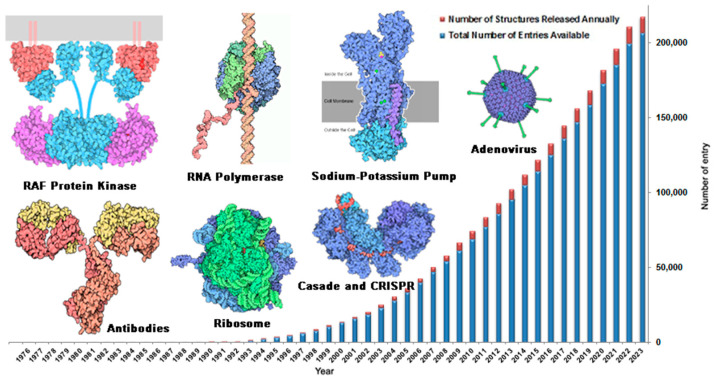
PDB statistics: overall growth of released structures per year (https://www.rcsb.org, accessed on 16 July 2023).

**Figure 4 ijms-24-13953-f004:**
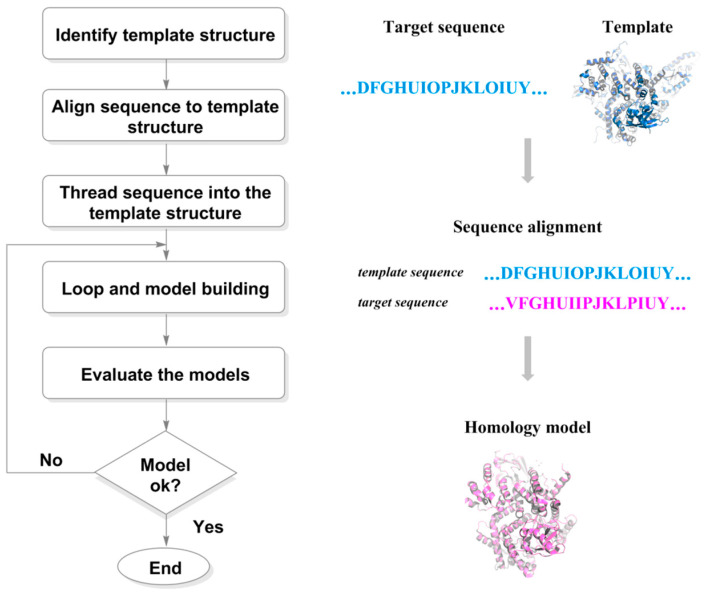
Steps involved in the homology model-building process.

**Figure 5 ijms-24-13953-f005:**
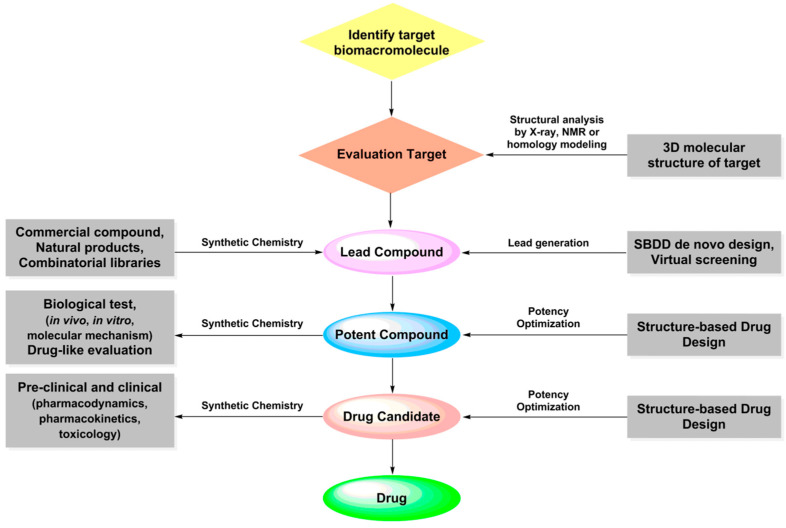
SBDD in drug discovery and design flow.

**Figure 6 ijms-24-13953-f006:**
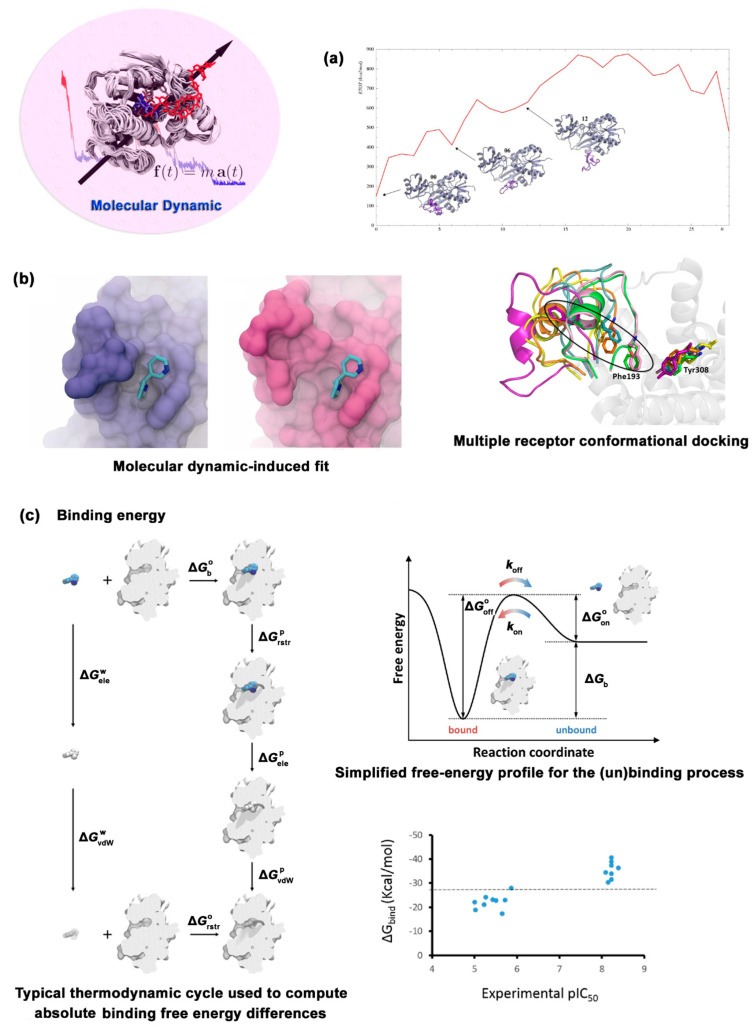
Three applications of MD in drug design. (**a**) Conformational simulation of flexible protein. (**b**) Refinement of ligand–target complex structure. (**c**) Calculation of binding energy. Figures are taken from refs. [81,86].

**Figure 7 ijms-24-13953-f007:**
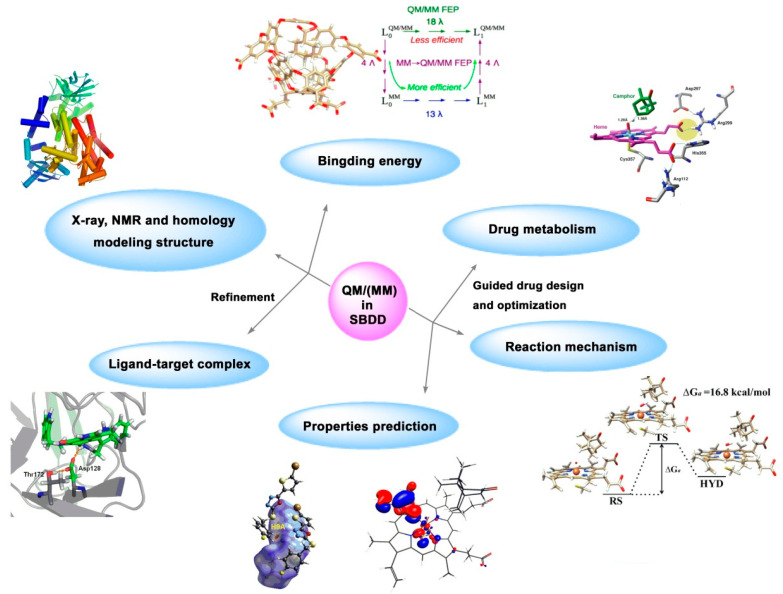
Application of QM and QM/MM in structure-based drug design.

**Figure 8 ijms-24-13953-f008:**
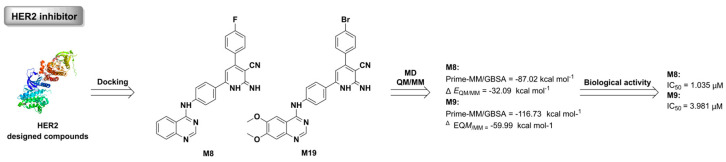
The discovery process for HER2 inhibitor using the molecular docking–MD–QM/MM approach.

**Figure 9 ijms-24-13953-f009:**
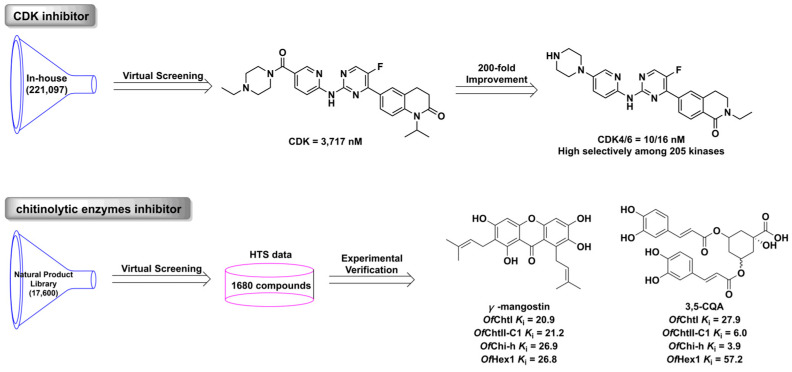
The discovery process for CDK and chitinolytic enzyme inhibitor using the virtual screening approach.

**Figure 10 ijms-24-13953-f010:**
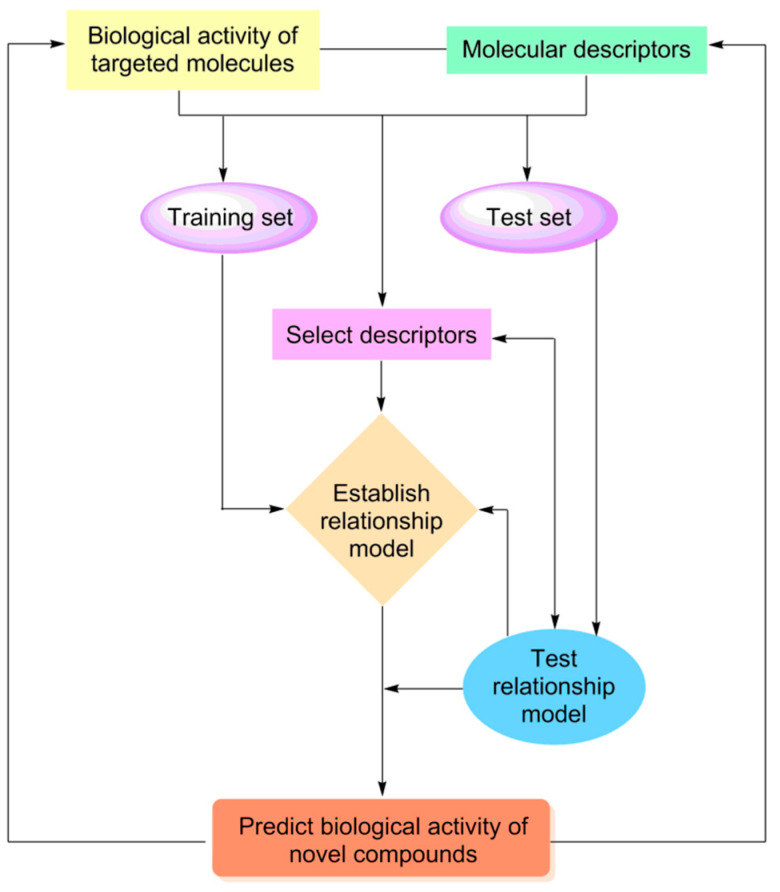
Flow diagram showing the investigation of the quantitative structure–activity relationship.

**Figure 11 ijms-24-13953-f011:**
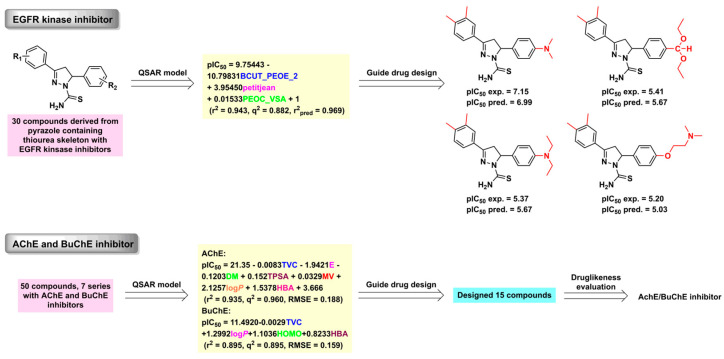
The discovery process for EGFR and AChE/BuChE inhibitors using the QSAR approach.

**Figure 12 ijms-24-13953-f012:**
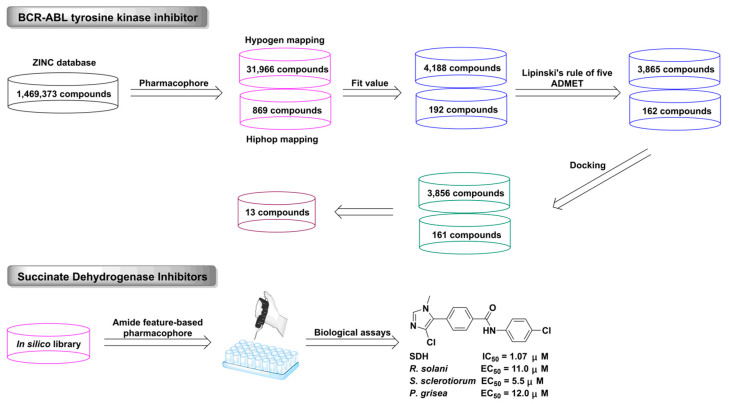
The discovery process for BCR-ABL tyrosine and succinate dehydrogenase inhibitors using the pharmacophore approach.

**Figure 13 ijms-24-13953-f013:**
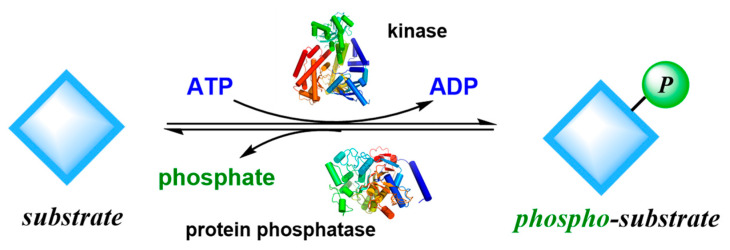
Phosphorylation and dephosphorylation processes are catalyzed by kinases and protein phosphatases.

**Figure 14 ijms-24-13953-f014:**
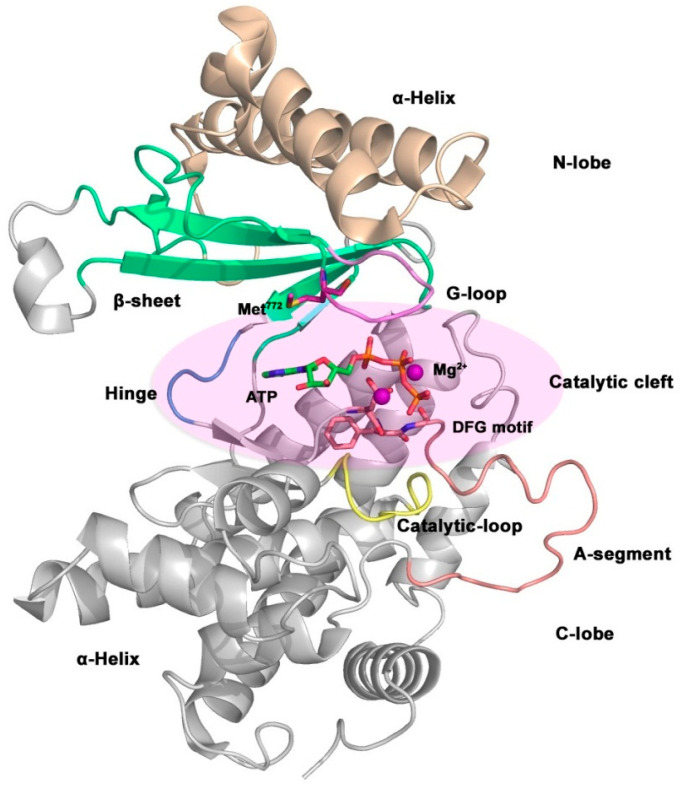
Overall crystal structure of the catalytic structure of the p110α subunit (PDB code: 1IRK).

**Figure 15 ijms-24-13953-f015:**
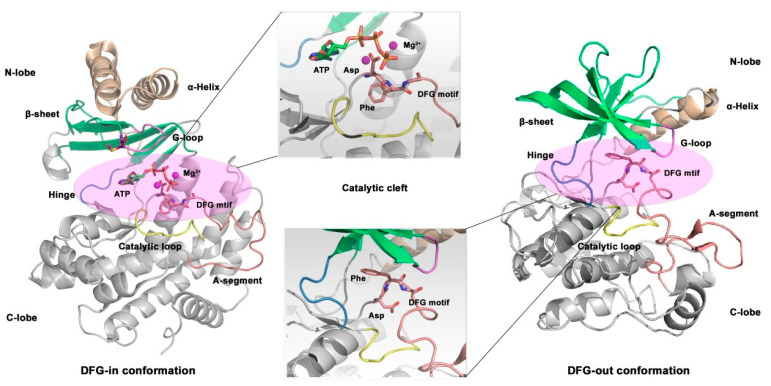
“DFG-in” and “DFG-out” conformations of p110α subunit (PDB code: 1IRK and 1E8X).

**Figure 16 ijms-24-13953-f016:**
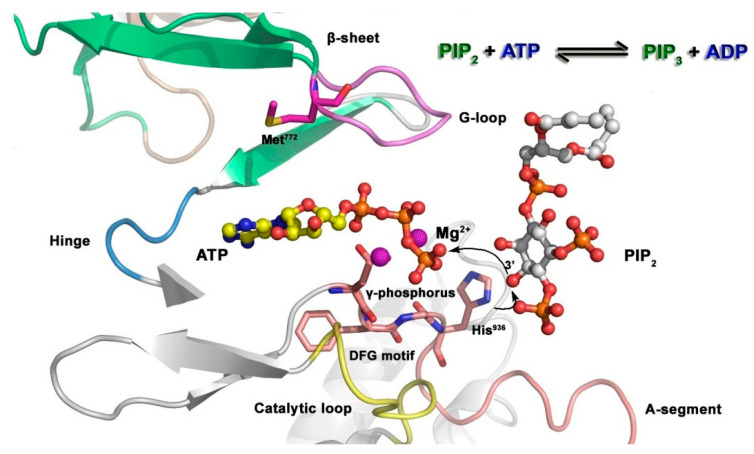
Mechanism of phosphorylation of p110α (PDB code: 1E8X). The oxygen molecule from the PIP_2_ substrate attacks the *γ*-phosphorus of ATP. His^936^ serves as a catalytic base by removing the proton from the hydroxyl group of PIP_2_.

**Figure 17 ijms-24-13953-f017:**
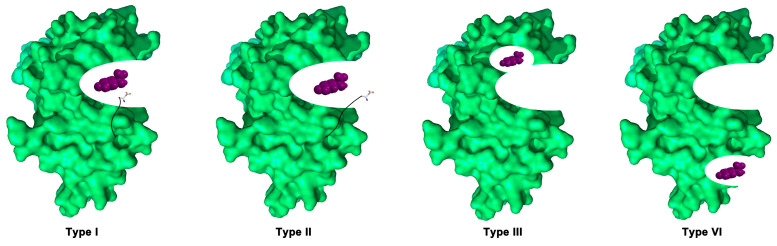
Four types of reversible binding modes. Type I inhibitors bind to the active conformation of the kinase, with the aspartate residue (white backbone) of the DFG motif pointing into the ATP-binding pocket; type II inhibitors bind and stabilize the inactive conformation of the kinase, with the flipped aspartate residue facing outward of the binding pocket; type III inhibitors occupy an allosteric pocket that is adjacent to the ATP-binding pocket but does not overlap with it; type IV inhibitors bind to an allosteric pocket remote from the ATP-binding pocket.

**Figure 18 ijms-24-13953-f018:**
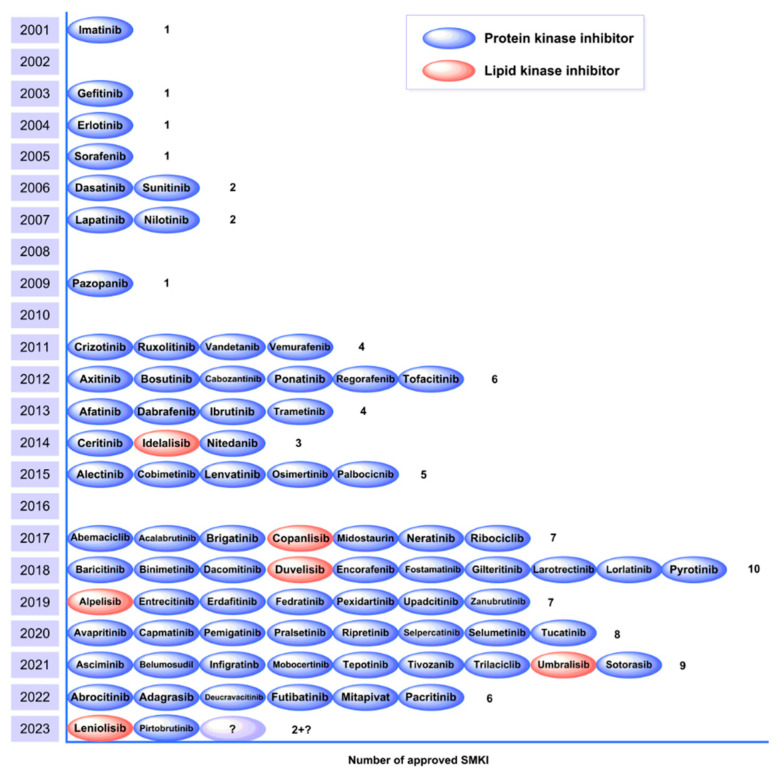
Number of approved SMKIs from 2001 to July 2023.

**Figure 19 ijms-24-13953-f019:**
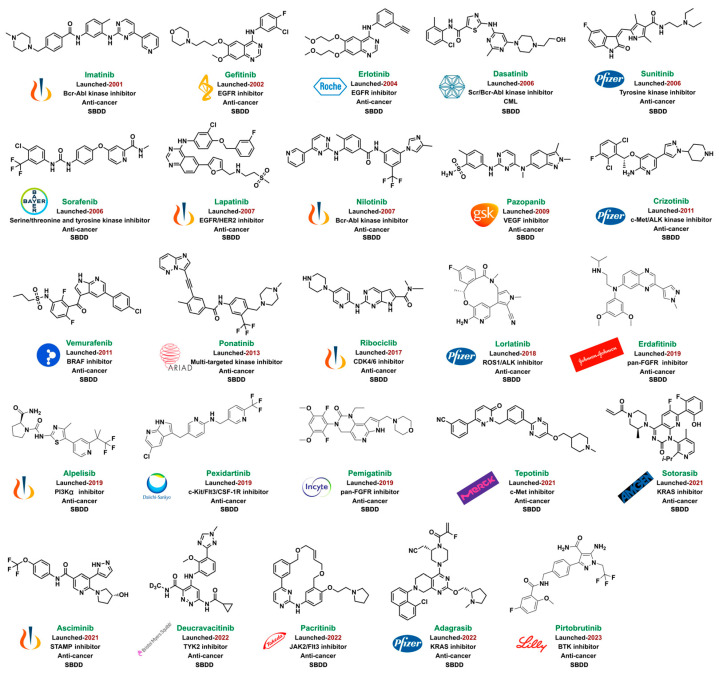
SMKIs discovered using CADD.

**Figure 20 ijms-24-13953-f020:**
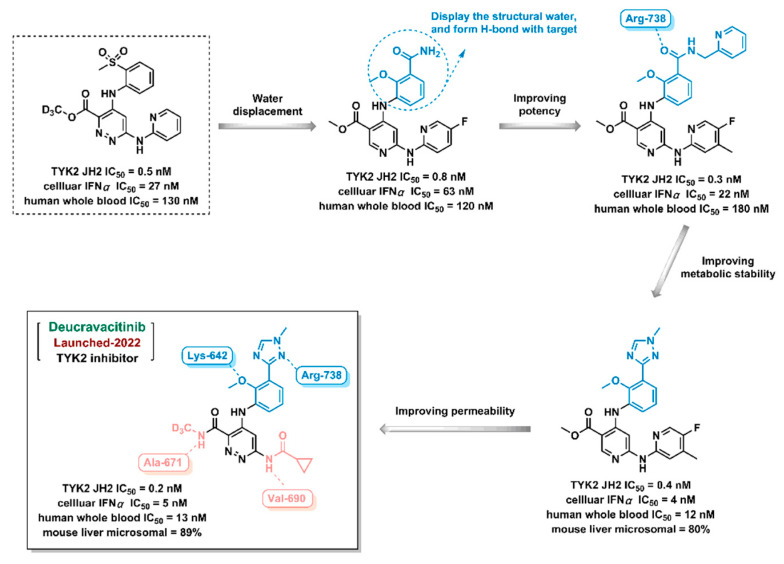
The Deucravacitinib discovery process using computer-aided drug design strategy.

**Figure 21 ijms-24-13953-f021:**
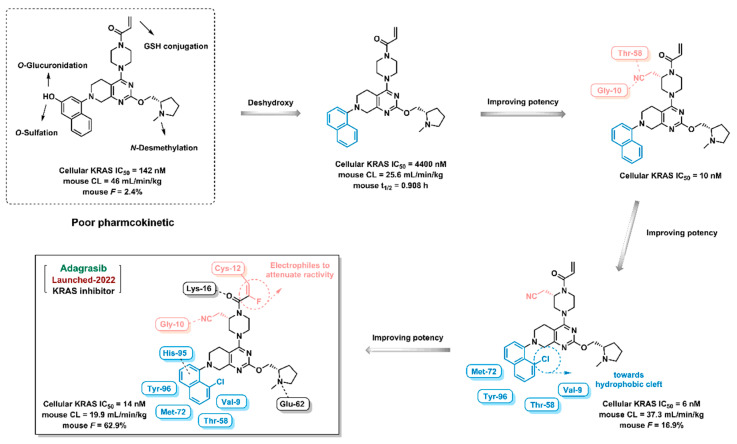
Adagrasib discovery process using computer-aided drug design strategy.

**Figure 22 ijms-24-13953-f022:**
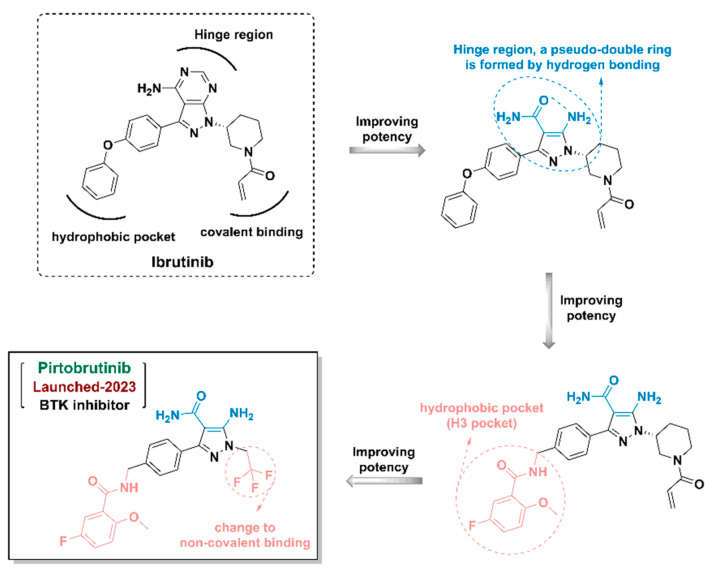
Pitobrutinib discovery process using computer-aided drug design strategy.

**Figure 23 ijms-24-13953-f023:**
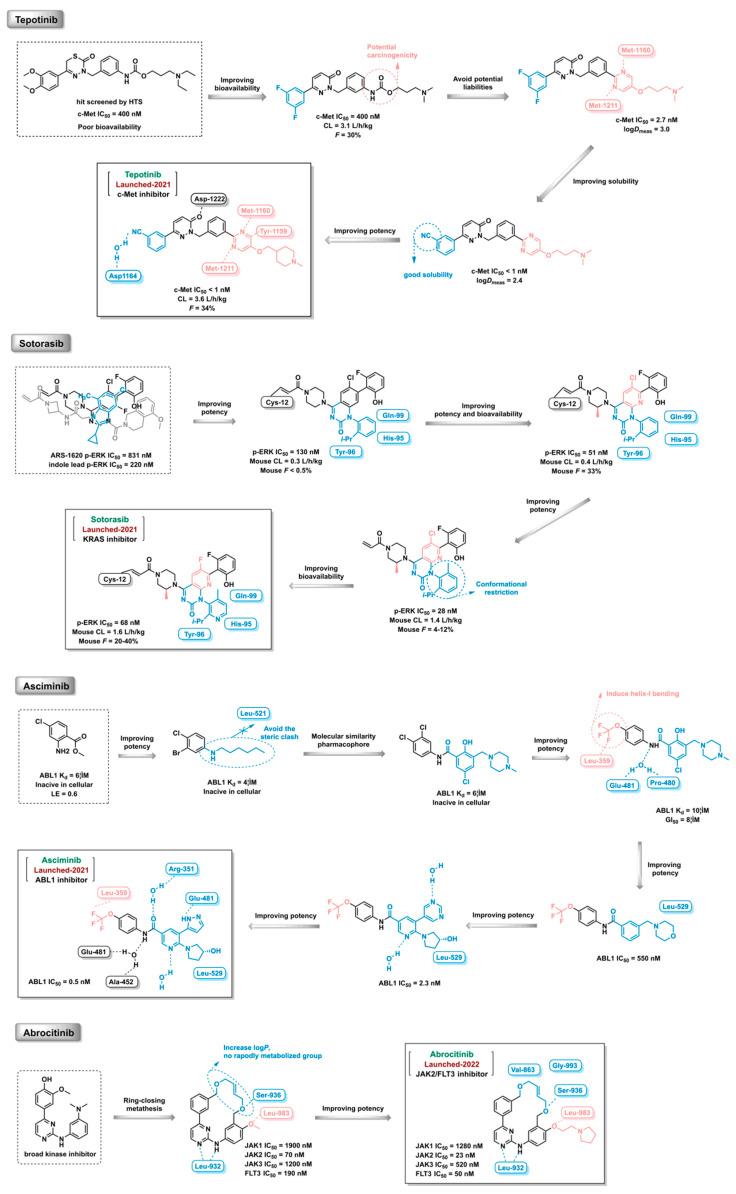
FDA-approved kinase inhibitors discovered using the computer-aided drug design strategy (2021–2023).

**Table 2 ijms-24-13953-t002:** Popular molecular dynamics software.

Software	Scoring Function	Charge	Website	Ref.
Amber	Mainly for biological system	AmberTools Free	http://ambermd.org/, accessed on 16 July 2023).	[87]
CPMD	Biological and chemical systems	Free	http://www.cpmd.org/, accessed on 16 July 2023).	[88]
NAMD	Biological and chemical systems	Free	http://www.ks.uiuc.edu/Research/namd, accessed on 16 July 2023).	[89]
Lammps	Material and solid-state physical systems	Free	https://www.lammps.org/, accessed on 16 July 2023).	[90]
Gromacs	Mainly for biological system	Free	https://www.gromacs.org/, accessed on 16 July 2023).	[91]
Charmm	Mainly for biological system	Free	https://www.charmm.org/, accessed on 16 July 2023).	[92]
Tinker	Mainly for biological system	Free	http://dasher.wustl.edu/tinker/, accessed on 16 July 2023).	[93]

**Table 3 ijms-24-13953-t003:** Quantum–chemical descriptions.

Definition	Name
	Charges
*Q* _A_	net atomic charge on atom A
*Q*_min_, *Q*_max_	net charges of the most negative and most positive atoms
*Q* _AB_	net group charge on atoms A and B
*Q*_T_, *Q*_A_	sum of absolute values of the charges of all the atoms in a given molecule
*Q*_T2_, *Q*_A2_	sum of squares of the charges of all the atoms in a given molecule or functional group
*Q* _m_	mean absolute atomic charge (i.e., the average of the absolute values of the charges on all atoms)
	HOMO and LUMO Energies
*E*_HOMO_, *E*_LUMO_	energies of the highest occupied molecular orbitals (HOMO) and lowest unoccupied molecular orbitals (LUMO)
∆*E*_LUMO-HOMO_	HOMO and LUMO orbital energy difference
*η* = (*E*_LUMO_ − *E*_HOMO_)/2	hardness
*S* = 1/(*E*_LUMO_ − *E*_HOMO_).	softness
∆*η* = *η*_R_ − *η*_T_	activation hardness. R and T stand for reactant and transition states
	Molecular Polarizabilities
*α*	molecular polarizability
*α* = (*α*_xx_ + *α*_yy_ + *α*_zz_)/3	mean polarizability of the molecule
*β*_2_ = [(*α*_xx_ − *α*_yy_)^2^ + (*α*_yy_ − *α*_zz_)^2^ + (*α*_zz_ − *α*_xx_)^2^]	anisotropy of the polarizability
	Dipole Moments and Polarity Indices
*µ*	molecular dipole moment
*µ*_char_, *µ*	charge and hybridization components of the dipole moment
*µ* ^2^	square of the molecular dipole moment
*D*_X_, *D*_Y_, *D*_Z_	components of dipole moment along inertia axes
*∆*	submolecular polarity parameter (largest difference in electron charges between two atoms)
*τ*	quadrupole moment tensor
	Energies
*E*	total energy
*H*	Enthalpy
*G*	Gibbs free energy
*S*	entropy
IP	ionization potential
EA	electron affinity, difference in total energy between the neutral and anion radical species
	Orbital Electron Densities
*q*_A_, *σ*, *q*_A_, *π*	*σ*- and *π*-electron densities of atom A
*Q*_A,H_, *Q*_A,L_	HOMO/LUMO electron densities of atom A
*F*_r_^E^ = *f*_r_^E^/*E*_HOMO_	electrophilic atomic frontier electron densities
*F*_r_^N^ = *f*_r_^N^/*E*_LUMO_	
	Atom–Atom Polarizabilities
*π*_AA_, *π*_AB_	self–atom polarizabilities and atom–atom polarizabilities
	Superdelocalizabilities
*S*_E, A_, *S*_N, A_	electrophilic and nucleophilic superdelocalizabilities

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
