# Peer review of "An Updated Review on Developing Small Molecule Kinase Inhibitors Using Computer-Aided Drug Design Approaches"

_ijms, 2023, doi:10.3390/ijms241813953_

Round 1

Reviewer 1 Report

Despite the significant achievements of specialists in the field of medicinal chemistry and pharmacology in the development of modern targeted low-toxic drugs for the treatment of socially significant human diseases, at present, due to the problem of the emergence of multiple drug resistance, the search for new effective medicines is an acute issue. In the process of searching for new structures, with a certain molecular target, molecular modeling methods are increasingly being used, which make it possible to predict the effectiveness of a particular structure with a certain degree of probability. The authors have prepared a good review on the application of CADD methods, such as docking, molecular dynamics, pharmacophore, for the development and optimization of small molecular weight kinase inhibitors. I think that the review will be of interest both to specialists in the field of organic, medicinal chemistry and pharmacology, and related specialties.

Meanwhile, I believe that the authors need to place greater emphasis on the universality of the methods described in the review, since they are suitable not only for the search for kinase inhibitors, but also, in conclusion, summarize all the material presented in the review by describing the most optimal algorithm for applying the complexes of the above approaches, as well as also review their main advantages and disadvantages.

Author Response

    We appreciated the constructive criticism and suggestion. We addressed all the points raised by the reviewers, as summarized below. It is noticeable that all changes in the revised manuscript, compared to the previous version, have been shown by Blue Highlights. The order of subtitles and references has been modified. In addition, after carefully proof-reading our paper, we figure out some spelling and grammatical errors. Therefore, we tried our best to improve the manuscript and made some changes. These changes would not influence the content and framework of the paper. We addressed all the points raised by the reviewers, as summarized below.

1. The authors need to place greater emphasis on the universality of the methods described in the review, since they are suitable not only for the search for kinase inhibitors, but also, in conclusion, summarize all the material presented in the review by describing the most optimal algorithm for applying the complexes of the above approaches, as well as also review their main advantages and disadvantages.

Authors’ Response: Thank you for your suggestion. We have summarized the application examples of each method in different types of drug development and listed them under the corresponding method entries, which marked by Blue Highlights. And, a flowchart was also presented for an example of drug design using this method (seen in Figure 8, 9, 10, 12). Furthermore, the advantages and disadvantages of different methods, as well as the optimal conditions for their application, are briefly described when introducing each method.

Reviewer 2 Report

The review entitled "An Updated Review of Developing Small Molecule Kinase Inhibitors Using Computer-Aided Drug Design Approaches" provides insights into recent advancements in CADD and SMKIs, focusing on structure-based and ligand-based drug design methods. It also emphasizes the progress in kinase inhibitors and their exploration through CADD. However, there are few points that has to be added before accepting it for publication:

It would be great to add one more column of references to the table 1 and table2.

2.        The title of the review is “An Updated Review of Developing Small Molecule Kinase Inhibitors Using Computer-Aided Drug Design Approaches”. I feel that the details about all the approaches applies in CADD has been very well described. However, there are very limited case studies have been mentioned in the review utilizing these approaches in terms of kinase drug discovery. Authors have only described it by using the example of three kinase inhibitors, Deucravacitinib, Adagrasib and Pirtobrutinib. It would be great to see how the different CADD approaches described in the paper (Molecular Docking, Molecular Dynamic, Quantum Chemistry methods for Structure-Based Drug Design; Virtual screening; Quantitative Structure-Activity Relationship, DFT-Based Quantitative Structure-Activity Relationship, Pharmacophore Modeling, Molecular Similarity methods for Ligand-Based Drug Design) have been utilized to identify inhibitors of different kinases. Authors can do this by adding few recent case studies either along with every section where they have described the approaches or as a separate section. This would do justice with the title of the review.

It would be also great to see a table listing different inhibitors identified by applying the various approaches of CADD as mentioned above. 

Author Response

    We appreciated the constructive criticism and suggestion. We addressed all the points raised by the reviewers, as summarized below. It is noticeable that all changes in the revised manuscript, compared to the previous version, have been shown by Blue Highlights. The order of subtitles and references has been modified. In addition, after carefully proof-reading our paper, we figure out some spelling and grammatical errors. Therefore, we tried our best to improve the manuscript and made some changes. These changes would not influence the content and framework of the paper. We addressed all the points raised by the reviewers, as summarized below.

1. It would be great to add one more column of references to the table 1 and table2.

Authors’ Response: The references of each software were added in Table 1 and Table 2.

2. It would be great to see how the different CADD approaches described in the paper (Molecular Docking, Molecular Dynamic, Quantum Chemistry methods for Structure-Based Drug Design; Virtual screening; Quantitative Structure-Activity Relationship, DFT-Based Quantitative Structure-Activity Relationship, Pharmacophore Modeling, Molecular Similarity methods for Ligand-Based Drug Design) have been utilized to identify inhibitors of different kinases. Authors can do this by adding few recent case studies either along with every section where they have described the approaches or as a separate section. This would do justice with the title of the review.

Authors’ Response: Thank you for your constructive comments. We have summarized the application examples of each method in different types of drug development and listed them under the corresponding method entries, which marked by Blue Highlights. And, a flowchart was also presented for an example of drug design using this method (seen in Figure 8, 9, 10, 12). Furthermore, the process of FDA-approved kinase inhibitors (2021-2023) discovered by computer-aided drug design strategy was descripted in Figure 23.

Round 2

Reviewer 2 Report

The authors have addresses all the concerned raised. The manuscript can be accepted in it's present form.